# Biological Control of Alternaria Leaf Spot Caused by *Alternaria* spp. in Jerusalem Artichoke (*Helianthus tuberosus* L.) under Two Fertilization Regimes

**DOI:** 10.3390/plants8110463

**Published:** 2019-10-30

**Authors:** Wanalai Viriyasuthee, Sanun Jogloy, Weerasak Saksirirat, Suwita Saepaisan, Mark L. Gleason, Ruey Shyang Chen

**Affiliations:** 1Department of Agronomy, Khon Kaen University, Khon Kaen 40002, Thailand; wanalai_v@hotmail.com; 2Department of Entomology and Plant Pathology, Khon Kaen University, Khon Kaen 40002, Thailand; weerasak@kku.ac.th (W.S.); suwitsa@kku.ac.th (S.S.); 3Department of Plant Pathology and Microbiology, Iowa State University, Ames, IA 50011, USA; mgleason@iastate.edu; 4Department of Biochemical Science and Technology, National Chiayi University, Chiayi 60004, Taiwan; rschen@mail.ncyu.edu.tw

**Keywords:** *Trichoderma harzianum*, integrate control disease, enzyme activity

## Abstract

The objectives of this study were to evaluate the efficacy of integrating resistant genotypes of Jerusalem artichoke with *Trichoderma harzianum* isolate T9 to control Alternaria leaf spot caused by *Alternaria* spp. under two fertilization regimes and to determine whether T9 application induced chitinase and β-1,3-glucanase activity in Jerusalem artichoke leaves. Six Jerusalem artichoke varieties (resistant varieties JA15, JA86, and JA116 and susceptible varieties HEL246, HEL293, and JA109) and three disease control methods (a non-inoculated control, application of *T. harzianum* T9, and fungicide sprays (propiconazole at a rate of 30 mL/20 L of water, 375 ppm)) was conducted in two separate trials (different fertilization regimes) at the experimental farm of the Faculty of Agriculture, Khon Kaen University, Khon Kaen, Thailand. Resistant genotypes controlled Alternaria leaf spot effectively. Application of *Trichoderma* showed low efficacy to control Alternaria leaf spot, but in specific susceptible genotypes—HEL246 and HEL293—the application of *Trichoderma* could reduce disease severity up to 10%. The application of *Trichoderma* was associated with a rise in production of chitinase and β-1,3-glucanase in HEL246 seedlings. The number of *Trichoderma* propagules in soil, as well as the extent of colonization of roots and leaves, were monitored. The results indicated that application of *Trichoderma* had higher propagules than non-inoculated control. Neither varietal resistance nor the disease control methods used in this study impacted the yield or yield components of Jerusalem artichoke.

## 1. Introduction

Jerusalem artichoke (*Helianthus tuberosus* L.), also known as sunchoke, a native of North America, can provide multiple benefits as a functional food and source of bioethanol and animal feed [1]. Tubers of Jerusalem artichoke produce large amounts of inulin, a long-chain fructose with a terminal glycopyranose unit at the reducing end [2]. Inulin has attracted human dietary interest as a prebiotic, since it confers sweetness but cannot be metabolized by the mammalian digestive system. Inulin promotes growth of beneficial microbes such as bifidobacterial and lactobacilli in the human gut, so it is considered to be a functional food [3]. Jerusalem artichoke is viewed as having potential as a new commercial crop suitable for production in tropical areas of the world [4].

A fungal leaf spot disease incited by an *Alternaria* spp. has emerged as a threat to Jerusalem artichoke production in Thailand [5]. Alternaria leaf spot can cause significant economic damage to a wide variety of additional hosts, including sunflower (*Helianthus* spp.), on which it can cause yield loss of up to 80% [6]. The conventional approach to control of Alternaria leaf spot—spray applications of synthetic chemical fungicides—has been recommended for potato (*Solanum tuberosum*) [7], chili pepper (*Capsicum annuum*) [8], sunflower [9], and Jerusalem artichoke [10].

However, many synthetic chemical fungicides have negative side effects, including environmental pollution, damage to human health, and loss of efficacy due to increased pathogen resistance to fungicides. Therefore, alternatives to chemical fungicides are being sought for Alternaria leaf spot management. For example, resistant genotypes have been recommended as a strategy for controlling this disease in peanut (*Arachis hypogea*) [11] and sunflower [12]. In addition, biocontrol methods against Alternaria leaf spot can provide alternatives to synthetic chemical fungicides that are much less damaging to people and the environment. *Trichoderma* includes several species that have been reported to be effective for control of Alternaria leaf spot in sunflower [13], aloe vera (*Aloe vera*) [14], peanut [15], and chili [16]. In peanut, *Trichoderma* inhibited *Alternaria alternata* mycelia growth in vitro and suppressed disease development index under field condition, but less effective than conventional fungicide control [15]. In Jerusalem artichoke, *T. harzianum* isolate T9 reduced incidence of stem rot (caused by *Sclerotium rolfsii*) by 30% in greenhouse trials [17] while increasing the activity of the fungal cell-wall-degrading enzymes chitinase and β-1,3-glucanase [18].

Plant nutrition may also play a role in Alternaria leaf spot control [19]. For example, cotton (*Gossypium hirsutum*) plants with potassium deficiency were more susceptible to *A. alternata* than those that were not deficient [20]. In potato, a low rate of nitrogen fertilization resulted in greater lesion area for potato early blight (caused by *Alternaria solani*) than a high nitrogen rate [21].

To our knowledge, alternatives to chemical fungicides for control of Alternaria leaf spot have not been investigated in Jerusalem artichoke but are worth exploring in order to mitigate environmental damage and reduce the risk of the development of fungicide resistance. The objectives of this study were to evaluate the efficacy of integrating resistant genotypes of Jerusalem artichoke with *T. harzianum* to control Alternaria leaf spot caused by *Alternaria* spp. under two fertilization regimes and to determine whether *T. harzianum* induces the activity of chitinase and β-1,3-glucanase in Jerusalem artichoke leaves.

## 2. Results

### 2.1. Disease Parameters

For combined analysis of variance, there was a significant difference of two-way interactions for genotypes × fertilization regimes for all disease parameters. Therefore, the results were presented separately for each fertilization regime. For individual analysis of variance of both fertilization regimes, the interaction of genotypes × disease control methods for all disease parameters were found. Therefore, the disease parameters of genotypes were presented in each disease control methods.

The *Trichoderma* and fungicide treatments showed low reduction in disease severity (0−9% and 0−10% for *Trichoderma* application and fungicide sprays, respectively), area under the disease progress curve for disease incidence (AUDPC-DI), and area under the disease progress curve for disease severity (AUDPC-DS) across all genotypes (Table 1). In the low-fertilization trial, HEL293 had the highest disease severity, AUDPC-DI, and AUDPC-DS under all three disease control methods, whereas JA86, JA116, and JA15 showed relatively low disease severity, AUDPC-DI, and AUDPC-DS.

Under the high-fertilization regime, the *Trichoderma* treatment, JA86 had the lowest disease incidence among the genotypes (Table 2). For disease control methods, the *Trichoderma* and fungicide treatments showed low reduction in disease severity (0−10% and 0−5% for *Trichoderma* application and fungicide sprays, respectively), AUDPC-DI, and AUDPC-DS across all genotypes. HEL246 and HEL293 had the highest disease severity, AUDPC-DI, and AUDPC-DS, whereas JA86, JA116, and JA15 showed the lowest disease severity, AUDPC-DI, and AUDPC-DS under all disease control methods.

### 2.2. Monitoring of Trichoderma Soil Populations and Plant Colonization

Under both fertilization regimes across genotypes, *Trichoderma* inoculation resulted in a higher number of *Trichoderma* propagules than non-inoculated control (data not shown). Among genotypes-plot, soil adjacent to HEL246 had the highest number of *Trichoderma* propagules. For root and leaf colonization, the *Trichoderma* treatment resulted in 99–100% incidence of colonization, whereas the non-inoculated control had no colonization (data not shown).

### 2.3. Enzyme Activity

#### 2.3.1. Chitinase Activity

For combined analysis of variance, significant differences for fertilization regimes × genotypes were observed for chitinase activity, therefore the results were presented in a separate fertilization regime.

The *Trichoderma* treatment had higher chitinase activity than the non-inoculated control treatment only at seven days after transferring seedling (DATS) under high fertilization regime (data not shown) but was lower at 15 days after transplanting (DAT) under a low fertilization regime. No significant difference was observed at 7 DATS and 30 DAT under a low fertilization regime and 15 and 30 DAT under a high fertilization regime (data not shown). For genotypes, HEL246 had the highest chitinase activity at 7 DATS and higher than non-inoculated control under both fertilization regimes (Figure 1a,b).

#### 2.3.2. β-1,3-Glucanase Activity

For combined analysis of variance, significant differences for fertilization regimes × genotypes were observed for β-1,3-glucanase activity, therefore the results were presented in a separate fertilization regime.

The *Trichoderma* treatment had higher β-1,3-glucanase activity than the non-inoculated control at 7 DATS under both fertilization regimes (data not shown), but lower at 30 DAT under high fertilization regime. No significant difference was observed at 15 and 30 DAT under low fertilization regime and 15 DAT under high fertilization regime across genotypes. AT 7 DATS, JA15 and HEL246 and JA109 had the highest β-1,3-glucanase activity, with the *Trichoderma* application under two fertilization regimes (Figure 1c,d) However, only HEL246 had higher β-1,3-glucanase activity than the non-inoculated control under both fertilization regimes.

### 2.4. Yield and Yield Components

No significant differences among disease control methods were found for tuber yield (Table 3) or yield components (number of tubers per plant and tuber size) across fertilization regimes (Table 4). For combined analysis of variance, significant differences were found for the interaction of genotypes × fertilization regime in tuber yield, so the results were presented in a two-way table.

Among genotypes, JA109 had the highest number of tubers per plant, whereas JA86 had the lowest number of tubers per plant (Table 4). JA86 had the largest tuber size, whereas JA116 and JA109 had the smallest tuber size. Under the low-fertilization regime, JA86 and HEL293 had the highest of tuber yield, whereas JA116 and JA109 had the lowest tuber yield under both fertilization regimes (Table 3).

## 3. Discussion

For the analysis of variance, the interaction between genotypes by disease control methods was observed. The results indicated that different efficacy of the control method depended on genotypes. For this study, applications of *Trichoderma* or fungicide sprays (propiconazole) suppressed disease severity, and AUDPC-DS of Alternaria leaf spot in Jerusalem artichoke genotypes under two fertilization regimes (Table 1 and Table 2). The different efficacy of *Trichoderma* to control the disease depended on level of resistant genotypes. Resistant genotypes, JA15, JA86, and JA116 [5] showed low disease severity (7–39% and 15–33% for low and high fertilization regimes, respectively), and AUDPC-DS in non-inoculated control, whereas the susceptible genotypes HEL246, HEL293, and JA109 [5] showed high disease severity (56–78% and 56–70% for low and high fertilization regimes, respectively). These findings supported that JA15, JA86, and JA116 had high resistance to Alternaria leaf spot. For high resistant genotypes, either application of *Trichoderma* or sprayed fungicide showed very low efficacy to reduce disease severity and AUDPC-DS under both fertilization regimes, so integrated control of resistant genotypes with *Trichoderma* or fungicide sprays (propiconazole) did not increase efficacy to control the disease. This outcome supports the conclusion that host plant resistance is the most practical, sustainable, and cost-efficient method for Alternaria leaf spot management in Jerusalem artichoke. For the susceptible genotypes, application of *Trichoderma* reduced the disease severity up to 10%, and AUDPC-DS only in HEL246 and HEL293 under both fertilization regimes. The results indicated low efficacy of *Trichoderma* application to control the disease. Similarly, *T. harzianum* was less effective to suppress Fusarium wilt in banana [22] and in tomato, only two of five isolates of *T. harzinum* inhibited growth of *S. rolfsii* according to in vitro and greenhouse trials [23]. Several contrasting reports include that *T. harzianum* reduced the disease incidence of Alternaria leaf spot more than 40% in bean [24], whereas in sunflower, *Trichoderma viride* reduced leaf spot severity caused by *Alternaria helianthi* more than 20% and increased seed yield [13]. The results of this study demonstrate the different efficacy of *Trichoderma* to control the disease depended on genotypes. Among susceptible genotypes, *Trichoderma* reduced disease severity only in genotypes HEL246 and HEL293 but did not reduce in JA109. Similar to previous reports, HEL246 was suppressed 20% of disease incidence and enhanced days to permanent wilting on stem rot caused by *S. rolfsii* after application of *T. harzianum* T9. In contrast JA37 found higher disease incidence and days to permanent wilting for application with *T. harzianum* T9 [17]. *Trichoderma* usually utilizes several mechanisms including antibiosis, competition, endophyte colonization, induced resistance, and mycoparasitism [25]. *Trichoderma* has two main components for biocontrol—direct activity of the plant pathogen by mycoparasitism and induced systemic resistance in plants [26]. In the present study, the hypothesis was *Trichoderma*-induced plant resistance and directly attacked the pathogen on Jerusalem artichoke. The results showed *Trichoderma* application had chitinase and β-1,3-glucanase activity higher than non-inoculated control only at 7 DATS in HEL 246 under two fertilization regimes, but there was no available data for HEL293. The chitinase and β-1,3-glucanase produced by plants defends against fungal infections in at least two different ways—by degrading fungal cell walls and promoting the release of cell-wall derived materials which can act as elicitors of defense reactions [27]. An upsurge of cell wall–degrading enzyme production is involved with plant resistance responses as pathogenesis-related proteins (PR protein). These responses are associated with a larger array of biochemical defenses against fungal pathogens [28]. Similarly, chitinase activity was also increased when inoculated with *Trichoderma* at five days after inoculation to control stem rot of Jerusalem artichoke in greenhouse conditions [18], whereas priming cucumber roots with *Trichoderma* spp. could increase activity of β-1,3-glucanase and chitinase up to 72 h post inoculation [29]. In the present study, at 15, and 30 DAT, chitinase and β-1,3-glucanase did not increase under both fertilization regimes, however, after inoculated with conidia, *Trichoderma* was able to survive in soil and colonize leaves and roots of Jerusalem artichoke. It might be because chitinases provide assets to enhance plant immunity and facilitate plant growth and development [30]. Either a less-effective strain of *T. harzianum* isolate T9 or inability to compete with native strains in the soil. These might be reasons for the low efficacy of *Trichoderma* to control Alternaria leaf spot in this study. For this study, the enzyme activity assay was a basic method to check plant expression, but further study should be carried out for a more accurate approach, such as gene expression using real time PCR. In this study, an enzyme activity assay was used as an index of expression of plant defense pathways. However, further studies are needed to characterize a wide range of defense pathway responses in Jerusalem artichoke against *Alternaria* spp.

Application of propiconazole decreased disease development in low efficacy of Jerusalem artichoke. In sunflower, severity of Alternaria leaf blight was reduced by applying propiconazole, and seed and oil yield increased [9]. Propiconazole also provided control of Alternaria leaf blight in peanut [15]. Despite its efficacy in controlling the disease in some crops, propiconazole was found to have several harmful effects on mammalian health [31]. Management tactics employing chemical fungicides should therefore be evaluated for impacts on applicators, consumers, and the environment, as well as for disease management.

Under the low-fertilization regime, most disease parameters were more severe than under the high-fertilization regime. In previous investigations, potassium promoted the development of thicker outer walls in epidermal cells, thus mitigating disease attack [32]. Similarly, the severity of Alternaria diseases in potato, tomato, and cotton was higher at lower levels of potassium [33]. Similar to potassium, the severity of disease was higher in the lower levels of calcium because it is essential for the functioning of plant membranes and as a component of plant cell wall structure [32]. In addition, plant tissues deficient in calcium are much more disease susceptible than non-deficient tissues [32]. Further study on the cost and benefit of *Trichoderma* and fungicide application for Alternaria leaf spot management in Jerusalem artichoke should be investigated in larger experimental plots across multiple seasons, sites, and years.

In the strategy for the management of Alternaria leaf spot, the high resistant varieties JA86 and JA116 should be used as a source of resistant lines and crossed with commercial varieties for high tuber yield, inulin content, and tuber quality. The segregating material should be used to select for resistant genotypes with high yield, inulin content, and quality for the grower.

## 4. Materials and Methods 

### 4.1. Experimental Design and Treatment

A field trial at the experimental farm (Khon Kaen University, Thailand) was arranged as a 6 × 3 factorial in randomized complete block design with four replications. Six Jerusalem artichoke varieties included three resistance against Alternaria leaf spot (JA15, JA86, and JA116) and three susceptible to the disease (HEL246, HEL293, and JA109) [5]. The three disease control methods included applications of *T. harzianum* (isolate T9), sprays of propiconazole fungicide at a rate of 30 mL/20L of water (375 ppm), and a control that received neither T9 nor propiconazole.

Normal cultural practices for Jerusalem artichoke in Thailand would specify the application of commercial fertilizer (e.g., 15-15-15 of N–P_2_O_5_–K_2_O at a rate of 312.25 kg ha^−1^). However, small farmers usually apply half that rate in order to cut input costs. The experiment was conducted under two fertilization regimes (as environments)—a low- fertilization regime (applied commercial fertilizer (15–15–15 of N–P_2_O_5_–K_2_O at a rate of 156.125 kg ha^−1^) and a high- fertilization regime (15–15–15 of N–P_2_O_5_–K_2_O at a rate of 312.25 kg ha^−1^) during October 2016–Febuary 2017. There was a 10-day difference in planting date between the two fertilization regimes in order to allow sufficient time for disease assessment in both plantings. Weather data at these sites during the testing period (maximum and minimum temperature, amount of rainfall, and relative humidity) in both fertilization regimes were recorded; results were similar under the two fertilization regimes.

### 4.2. Preparation of T. harzianum T9 Inoculum

*T. harzianum* isolate T9, provided by the Department of Entomology and Plant Pathology, Khon Kaen University, Khon Kaen, Thailand, was isolated from soil at Nong Khai, Thailand [18] cultured on potato dextrose agar (PDA) and incubated at room temperature (28 ± 2 °C). Five days after incubation, 0.5-mm-diameter plugs of *T. harzianum* colony were excised by a cork borer and transferred to plastic bags containing 100 g of steam-sterilized sorghum (*Sorghum bicolor*) seeds. After three days of incubation at room temperature (28 ± 2 °C), the bags were shaken gently to separate individual seed and ensure thorough colonization by *T. harzianum*. The mixture was then incubated for 10 days until a layer of green spores covered all seeds.

For preparation of a *T. harzianum* spore suspension, the *Trichoderma*-infested seed was suspended in sterile water and then filtered through a layer of sterilized muslin cloth. The spore concentration was diluted to 10^6^ conidia/mL with the aid of a hemocytometer.

### 4.3. Plant Material Preparation and Experimental Management

Soil preparation began by plowing three times, using conventional tillage equipment. Chemical fertilizer (15–15–15 of N–P_2_O_5_–K_2_O) was applied 30 DAT. Soil samples were taken at 0–30 cm depth by auger at three days before transplanting and at 37 DAT [34]. At each soil sampling time, samples were mixed thoroughly to assure uniformity and air-dried. The soil was assessed for soil texture, pH, organic matter [35], total N [36], available P [37], exchangeable K, exchangeable Ca, cation exchange capacity, and electrical conductivity. Raised beds (1.6 × 3 × 0.5 m) were created by tillage; spacing between plot-bed was 1.5 m. After application of fertilizer, the soil chemical and physical properties were similar, except for exchangeable K and Ca as follows: exchangeable potassium was 21.73 and 25.60 mg/kg, and exchangeable calcium was 325 and 340 mg/kg for low and high fertilization regimes, respectively.

Jerusalem artichoke genotypes were obtained from the Plant Gene Resource of Canada, Saskatoon, SK, Canada (JA15, JA86, JA116, and JA109) and the Leibniz Institute of Plant Genetics and Crop Plant Research, Gatersleben, Germany (HEL 246 and HEL293). Tubers were cut into small pieces with two to three active buds each and incubated for one week in moist coconut coir dust to facilitate uniform germination. The germinated tuber pieces were then transferred to plug trays with a mixture of soil and charred rice husks (1:1) until each seedling had two leaves. For the *T. harzianum* treatment, the medium was thoroughly mixed with 10 g of *Trichoderma*-infested seed inoculum per 1 kg medium. Healthy-appearing seedlings at the two-leaf stage were transplanted to field plots. Plant spacing was 0.4 m between rows and 0.3 m between plants within rows. For the *T. harzianum* treatment, three days before transplanting, *Trichoderma*-infested seed inoculum was incorporated into the soil at a rate of 200 g of the inoculum per plot. In addition, a spore suspension of T9 (10^6^ conidia/ ml) was sprayed at a rate 10 mL/ plant at 20 and 45 DAT. For the fungicide treatment, propiconazole (Conacide, Nonthaburi, Thailand) at a rate of 30 mL/20 L of water (375 ppm) was sprayed at 30 DAT [9]. Hand weeding was performed at 27 and 60 DAT. Mini-sprinkler irrigation was available as necessary to avoid drought stress.

### 4.4. Monitoring of T. harzianum Propagules in Soil and Colonization of Roots and Leaves

*T. harzianum* populations in soil and colonization of roots and leaves were assessed in plots of JA15, JA116, HEL246, and JA109, both for the non-inoculated control and *T. harzianum* application treatments. Soil samples were collected from both treatments on the day of transplanting. The soil was collected by auger for 30 cm depth at five points per subplot and mixed well. The soil samples were subsampled for making soil dilution spread plates [38]. One gram of soil was placed in a test tube containing 9 mL of sterile distilled water and 10-fold serially diluted with sterile distilled water were made up to 10^−7^. Aliquots of 100 µL were taken from each dilution and spread on Martin’s medium [39]. The spreader plates were incubated at room temperature (28 ± 2 °C) for 24 h. After incubation, colonies of *T. harzianum* were counted on each plate. These numbers were recalculated to represent the number of *T. harzianum* T9 propagules in 1 g of soil as follows:
CFU/g soil = (A × B × C)/(D × E)(1)
where A is the number of *T. harzianum* T9 colonies, B is the volume of added sterilized water (ml), C is a dilution factor, D is the weight of the soil sub-sample (g), and E is the volume of spreading the soil solution (mL).

Root samples were taken from *T. harzianum*-inoculated and non-inoculated control treatments 7 DATS. Roots of a single Jerusalem artichoke plant from each subplot were washed in sterilized water to remove media, then cut into 0.5 cm–long sections from root tip up to 3 cm, washed in sterile distilled water and dried with sterilized paper towels. Ten pieces of root were randomly placed on Martin’s medium plate and incubated at room temperature (28 ± 2 °C) for three days. The percentage for pieces of root colonization by *T. harzianum* was calculated as follows:
% pieces of colonized root = (number of roots colonized by *Trichoderma*/total number of roots) × 100(2)

Colonized leaves of Jerusalem artichoke were examined at 30 DAT from *T. harzianum*-inoculated and non-inoculated control treatments. Two leaves from each plot were sampled. After that, the leaves were cut into 0.5 cm–long from the leaf blade, avoiding the midrib. Five pieces per leaf were washed in sterile distilled water and dried with sterilized paper towels. The method used was the same as described previously for root colonization.

### 4.5. Enzyme Activity Analysis

Leaf samples of Jerusalem artichoke were collected from JA15, JA116, HEL246, and JA109 in non-inoculated control and *T. harzianum*-inoculated treatments at 7, 15, and 30 DAT. Different plants in each subplot were sampled on each sampling date, so that no plant was sampled more than once. Six leaves from the top, medium, and low layer of plant from two plants were harvested from each subplot, placed in a plastic ziplock bag, and placed in an insulated cooler on ice for transport to the laboratory, where they were stored at −20 °C for crude protein extraction. Leaves were extracted for crude protein and analyzed for the activity of chitinase and β-1,3-glucanase. Samples (0.5 g) were ground in liquid nitrogen by using a mortar, then amended with 1200 μL of 0.1 M sodium acetate buffer pH 5.0 (sodium acetate trihydrate 13.6 g/ H_2_O 1-L adjusting pH levels by 1 N acetic acid). Samples were centrifuged at 4 °C and 14,000 rpm for 20 min. The supernatant was stored at −20 °C until analysis. Protein concentration analysis followed the Bradford method [40], using bovine serum albumin (BSA) as a standard protein. 

For the chitinase activity assay, the reaction mixture was 0.1 mL of crude protein and 0.9 mL of 1% colloidal chitin used as a substrate in 0.1 M acetate buffer pH 5.0. The mixtures were incubated at 37 °C for 1 h, after which the reaction was stopped by heating at 100 °C for 20 min [41]. Reducing sugar in the supernatant was determined by spectrophotometry according to the method of Somogyi [42]. *N*-acetylglucosamine equivalent was evaluated as a measure of chitinase activity. One unit of enzyme activity was equated with 1 μmol of *N*-acetylglucosamine equivalent released from colloidal chitin within one hour per mg of protein in the reaction mixture.

Activity of β-1,3-glucanase was evaluated by incubation in 0.09% laminarin (β-1,3-glucan, Sigma Co., St. Louis, MO, USA) solution in 0.1 M acetate buffer, pH 5.0 in a volume of 0.9 mL with 0.1 mL of crude protein at 37 °C for 1 h. The reaction was stopped by heating at 100 °C for 20 min [41]. The reducing sugar (glucose equivalent) released from laminarin was reduced for enzyme analysis following the method of Somogyi [42]. One unit of enzyme activity was evaluated by the amount of glucose equivalent released from laminarin within 1 h per mg of protein in the reaction mixture.

### 4.6. Disease Assessment

Disease assessment was performed at three-day intervals from 20 to 81 DAT. Twelve plants per subplot, except bored row plants, plants for enzyme activity analysis, and plants for monitoring of *T. harzianum* colonization of roots and leaves, were assessed individually for the number of symptomatic plants and disease score (an index of disease severity). Disease score was determined by the qualitative rating scale developed by Mayee and Datar [43]. 

Disease incidence (DI) was calculated as follows [44]:DI (%) = (number of symptomatic plants/total number of plants) × 100(3)

Disease severity (DS) was calculated as follows [44]:DS (%) = (∑ [(rating score × number of plants in rating) × 100]/(total number of sampled plants × highest rating)(4)

The area under the disease progress curve (AUDPC) was calculated for disease incidence (AUDPC-DI) and disease severity (AUDPC-DS) using the formula provided by Ojiambo et al. [45].
AUDPC = ∑_i=1_[(X_i_ + X_i+3_)/2] × (t_i+3_ − t_i_)(5)
where x_i_ is disease incidence or disease severity on day i, x_i+3_ is disease incidence or disease severity on day i + 3, t_i_ is disease incidence or disease severity assessment on day i, and t_i+3_ is disease incidence or disease severity assessment on day i and i+3.

### 4.7. Yield and Yield Components of Jerusalem Artichoke

Five plants in each subplot, excluding border row plants, were sampled at maturity and used for the determination of yield components, number of tubers per plant, and tuber size. The number of tubers per plant was counted and averaged. Fresh tuber weight from five plants was determined and then divided by the number of tubers to obtain average tuber size (g/tuber). 

Fresh weight of tubers was determined from 12 plants in each plot. Tubers were harvested, washed, and weighed, then the weights were divided by number of harvested plants to determine tuber yield (g/plant).

### 4.8. Statistical Analysis

Data set of each fertilization regime (environment) were analyzed according to analysis of variance. Error variances of individual analysis were tested for homogeneity. Data sets that complied with homogeneity of error variance were used for combined analysis. The results from combined analysis of variance showed that the genotype × fertilization regimes were significant for all disease parameters. The results indicated that the expression of genotypes and efficacy of disease control methods were different in the two fertilizer regimes, so the expression of genotypes and efficacy disease control methods were presented in each fertilization regimes base on individual analysis [46]. The least significant difference was used to compare treatment means [47]. Disease incidence and disease severity at 77 days after transplanting were presented because the data showed the highest F-test discriminates among genotypes and the lowest CV value. All calculations were done using the STATISTIX8 (Analytical Software, Tallahassee, FL, USA) software package [48]. 

## 5. Conclusions

In conclusion, resistant genotypes of Jerusalem artichoke alone showed low disease severity, AUDPC-DI, and AUDPC-DS and similar efficacy for suppression of Alternaria leaf spot as integrate control of resistant varieties and *Trichoderma* application or propiconazole sprays. Application of *T. harzianum* T9 or propiconazole sprays showed low reduction of disease severity (10% or less), and AUDPC-DS of Alternaria leaf spot in susceptible genotypes, especially HEL246 and HEL293, under two fertilization regimes. The application of *T. harzianum* induced an increase of chitinase and β-1,3- glucanase only at seedling stage in HEL246. Neither host resistance nor the biocontrol or fungicide strategies impacted yield or yield components of Jerusalem artichoke.

## Figures and Tables

**Figure 1 plants-08-00463-f001:**
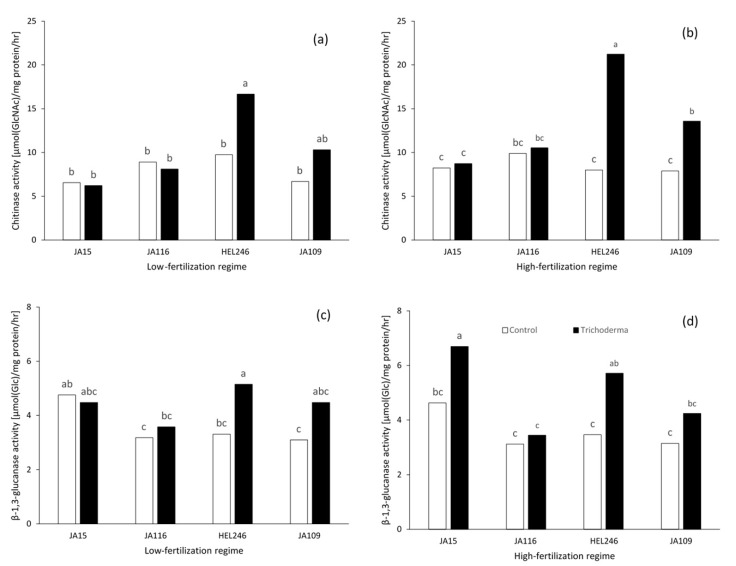
Enzyme activity of *Trichoderma* application and non-*Trichoderma* application for four Jerusalem artichoke genotypes at seven days after transferring seedling (DATS) under two fertilization regimes. (**a**) Chitinase activity at low fertilization regime; (**b**) Chitinase activity at high fertilization regime; (**c**) β-1,3-glucanase activity at low fertilization regime; (**d**) β-1,3-glucanase activity at high fertilization regime. Different letters indicate significant differences at *p* ≤ 0.05 among treatments.

**Table 1 plants-08-00463-t001:** Means of disease parameters under low- fertilization regime of six Jerusalem artichoke genotypes and three control methods.

**Genotypes**	**Disease Incidence (%) ^1^**					**Disease Severity (%) ^1^**			
	**Non-Inoculated Control**	***Trichoderma***	**Fungicide**	**Mean**		**Non-Inoculated Control**	***Trichoderma***	**Fungicide**	**Mean**
JA15	100 a	100 a	100 a	100 A		39 d	33 e	33 e	35 D
JA86	67 b	0 c	50 b	39 B		7 fg	0 h	6 g	4 F
JA116	100 a	100 a	100 a	100 A		11 f	11 f	11 f	11 E
HEL246	100 a	100 a	100 a	100 A		78 a	69 b	68 b	72 B
HEL293	100 a	100 a	100 a	100 A		78 a	75 a	75 a	76 A
JA109	100 a	100 a	100 a	100 A		56 c	56 c	56 c	56 C
Mean	94 A	83 B	92 A			45 A	41 B	42 B	
**Genotypes**	**Areas under Disease Progress Curve for Disease Incidence**					**Areas under Disease Progress Curve for Disease Severity**			
	**Non-Inoculated Control**	***Trichoderma***	**Fungicide**	**Mean**		**Non-Inoculated Control**	***Trichoderma***	**Fungicide**	**Mean**
JA15	2606 c	2575 c	2550 c	2577 C		485 e	453 e	448 e	462 C
JA86	413 ef	0 g	230 fg	214 E		46 f	0 f	26 f	24 D
JA116	700 d	525 de	579 de	601 D		78 f	58 f	64 f	67 D
HEL246	5506 a	5528 a	5416 a	5483 A		1840 a	1534 bc	1501 c	1625 A
HEL293	5581 a	5559 a	5341 a	5494 A		1811 a	1523 c	1609 b	1648 A
JA109	3950 b	3944 b	3769 b	3888 B		1189 d	1160 d	1138 d	1163 B
Mean	3126 A	3022 B	2981 B			908 A	788 B	798 B	

Means followed by the same small letters were not significantly different treatment in combination according to least significant difference (LSD) at *p* ≤ 0.05. Means followed by the same capital letters were not significantly different for each of main effects, according to least significant difference (LSD) at *p* ≤ 0.05. ^1^ Disease incidence and disease severity at 77 days after transplanting.

**Table 2 plants-08-00463-t002:** Means of disease parameters under high- fertilization regime of six Jerusalem artichoke genotypes and three control methods.

**Genotypes**	**Disease Incidence (%) ^1^**					**Disease Severity (%) ^1^**			
	**Non-Inoculated Control**	***Trichoderma***	**Fungicide**	**Mean**		**Non-Inoculated Control**	***Trichoderma***	**Fungicide**	**Mean**
JA15	100 a	100 a	100 a	100 A		33 e	33 e	33 e	33 C
JA86	100 a	63 b	100 a	88 B		15 f	7 h	11 g	11 E
JA116	100 a	100 a	100 a	100 A		16 f	11 g	11 g	13 D
HEL246	100 a	100 a	100 a	100 A		70 a	64 bc	66 b	67 A
HEL293	100 a	100 a	100 a	100 A		71 a	61 c	66 b	66 A
JA109	100 a	100 a	100 a	100 A		56 d	56 d	56 d	56 B
Mean	100 A	94 B	100 A			44 A	39 C	40 B	
**Genotypes**	**Areas under Disease Progress Curve for Disease Incidence**					**Areas under Disease Progress Curve for Disease Severity**			
	**Non-Inoculated Control**	***Trichoderma***	**Fungicide**	**Mean**		**Non-Inoculated Control**	***Trichoderma***	**Fungicide**	**Mean**
JA15	2150 f	2006 f	2150 f	2102 D		402 f	374 f	397 f	391 D
JA86	820 g	225 i	641 h	562 E		97 g	25 g	71 g	64 E
JA116	627 h	519 h	532 h	559 E		77 g	58 g	59 g	65 E
HEL246	5469 a	5444 ab	5281 b	5398 A		1864 a	1453 bc	1576 b	1631 A
HEL293	5453 ab	5459 a	4958 c	5290 B		1801 a	1383 c	1410 c	1532 B
JA109	4213 d	3956 e	3981 e	4050 C		1208 d	1067 e	1050 e	1108 C
Mean	3122 A	2935 B	2924 B			908 A	727 B	761 B	

Means followed by the same small letters were not significantly different treatment in combination according to least significant difference (LSD) at *p* ≤ 0.05. Means followed by the same capital letters were not significantly different for each of main effects, according to least significant difference (LSD) at *p* ≤ 0.05. ^1^ Disease incidence and disease severity at 77 days after transplanting.

**Table 3 plants-08-00463-t003:** Means of tuber yield (g/plant) under two fertilization regimes of six Jerusalem artichoke genotypes and three control methods.

Factors	Low-Fertilization Regime			High-Fertilization Regime
**Genotypes**				
JA15	182 bc			273 b
JA86	243 a			278 ab
JA116	107 d			150 c
HEL246	189 bc			330 a
HEL293	222 ab			321 ab
JA109	150 cd			171 c
**Treatments**	**Across Fertilization Regimes**			
Control (non-inoculated)			213 a	
*Trichoderma*			219 a	
Fungicide			221 a	

Means followed by the same small letters in the same column were not significantly different according to least significant difference (LSD) at *p* ≤ 0.05.

**Table 4 plants-08-00463-t004:** Means of the number of tubers per plant and tuber size (g/tuber) averaged over two fertilization regimes of six Jerusalem artichoke genotypes and three control methods.

Factors	Number of Tubers/Plant		Tuber Size (g/tuber)
**Genotypes**			
JA15	15.7 b		14.4 b
JA86	8.9 c		30.2 a
JA116	16.5 b		7.7 c
HEL246	16.3 b		15.8 b
HEL293	17.6 b		15.4 b
JA109	20.7 a		7.8 c
**Treatments**			
Control (non-inoculated)	16.3 a		14.5 a
*Trichoderma*	15.0 a		15.9 a
Fungicide	16.5 a		15.3 a

Means followed by the same small letters in the same column were not significantly different according to least significant difference (LSD) at *p* ≤ 0.05.

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
