# Peer review of "Biological Control of Alternaria Leaf Spot Caused by Alternaria spp. in Jerusalem Artichoke (Helianthus tuberosus L.) under Two Fertilization Regimes"

_plants, 2019, doi:10.3390/plants8110463_

Round 1
Reviewer 1 Report
The manuscript has been adequately improved and I support its publication
The scientific soundness is very good since the authors describe a well designed experimental plan that was also very well executed. The work presented is novel and could be of use to both practitioners as well as researchers. The English has been significantly improved at the submitted revision and is at good standing in rev2 text. The overall merit is high since the objectives of this study were to evaluate the efficacy of integrating resistant genotypes of Jerusalem artichoke with T. harzianum to control Alternaria leaf spot under two fertilization regimes and to determine whether T. harzianum induces the activity of chitinase and β-1,3-glucanase activity in Jerusalem artichoke leaves. Although, neither varietal resistance nor the disease control methods used in this study impacted yield or yield components of Jerusalem artichoke, resistant genotypes controlled Alternaria leaf spot effectively. Application of Trichoderma showed low efficacy to control Alternaria leaf spot, however, in specific susceptible genotypes, the application of Trichoderma could reduce disease severity up to 10%.
Author Response
October 23, 2019
Dear Editor/Reviewers
We are pleased to resubmit the manuscript no. Plants-614825 entitled “Biological Control of Alternaria Leaf Spot Caused by Alternaria spp. in Jerusalem artichoke (Helianthus tuberosus L.) under Two Fertilization Regimes” after revision for re-evaluation and possible publication in the Plants.
The authors would like to thank the editor and reviewers for all valuable comments and suggestion on the manuscript in order to improve the manuscript.
We have revised the manuscript based on editor and reviewers, comments and suggestions. All of the points and responses from the editor and reviewers done and attached with this file.
Best Regards,
Sanun Jogloy

Reviewer 2 Report
I have listed some changes that are not significant to the overall paper but would help with consistency of format and for correctness of presentation. However, the Discussion section changes are necessary. They can be changed in an alternative way to that which I suggest, but changes are needed for clarity of language:
Line 23 - Second parenthesis is not needed after 375 ppm.
Line 62 - Space needed before [17]
Line 92 - Space needed after 5%
Line 176 - Remove the words "the" and "a"
Line 178 - Remove the word "of"
Line 184 - Remove the word "that"
Line 185 - Rewrite as "Similarly, T. harzianum was less effective..."
Line 186 - Rewrite as "...only 2 out of 5 isolates of T. harzianum inhibited..."
Line 187 - Replace "conditions" with "trials".
Line 187/188 - Rewrite as "Several contrasting reports include, in bean,..."
Line 190/191 - I don't know how to change this sentence - but it carries no meaning as it is written.
Line 193 - Rewrite as "Similar to previous reports ,..."
Line 198 - Rewrite as "...biocontrol; direct activity....
Line 200 - Replace "with" with the word "the".
Line 206 - Add a period after (PR protein).
Line 212 - Rewrite as "...inoculated with conidia"
Line 214/216 - Rewrite as "Either a less-effective strain of T. harzianum isolate T9 or inability to compete with native strains in the soil, these might be reasons for low efficacy..."
Line 217 - Rewrite as "...assay was a basic method to check plant expression, however, a further study..."
Line 218 - Add a coma after the word "approach".
Line 220 - Rewrite as "...further studies are needed..."
Line 240 - Rewrite as "...used as a source of resistant lines and crossed with commercial varieties for high tuber yield..."
Line 251 - Remove the word "Following"
Line 260 - After "regimes." a new line should start
Line 332 - Remove space(s) before the word "sodium"
Line 385 - Add the word "an" after the word "induced"
Line 438 - Italicize "Trichoderma"
Author Response

(The authors gave the same response as above.)

Reviewer 3 Report
General comments
The results are interesting but the quality of the text and also the quality of the graphs has to be improved. It is indicated that the experimental design was a factorial between two factors (6 varieties x 3 control methods), but in fact two levels of fertilizer were also applied; it could be analyzed like a (6 x 3 x 2). How do you justify the choice you did ? What about the block effect (significant or not) ?
Specific comments
The abstract should be improved. The objectives you indicated at the beginning are not complete. You studied also the fungicide effect. The introduction is short but well done.
The part Results must be improved.
Lines 77-82: I suppose you did a 2-way analysis of variance (genotype and fertilization effects) with interaction, and with a block effect. You indicated that you obtained a significant effect for the interaction “genotypes x fertilization” but the results were not presented. The value of the interaction was around the values of the main effects or inferior? Why did you choose this analysis over a 2 way-analysis of variance (disease control x fertilization) and especially a three-way analysis of variance (genotypes x disease control x fertilization). If you did this choice you have to explain why.
Lines 119-120: You indicated also that you obtained an interaction (significant?), it could be interesting to have more details. (same thing lines 129-131).
There are several problems with the English, for instance line 151: “…significant differences were found in for the interaction ….” .
The part Discussion must also be improved.
Line 167: You began the discussion with a result not shown. So, you indicated again that the interaction ‘genotypes x disease control) was observed, but was it significant?
In the Materials and Methods part, the “Statistical analysis” has to be better explained, with more details and justifications.
In conclusion, the manuscript would need a lot of improvements before publication.
Round 2
Reviewer 3 Report
The text was improved, but the part 4.8 Statistical analysis needs still improvement (not clear)
it could be: As there was a significant interaction (genotype x fertilization) for all disease parameters, the data for each fertilization regime were analyzed using a two-way analysis of variance (variety and disease control) in a randomized complete block design. The homogeneity of the error differences between the two fertilization regimes was tested.
for the 1st part (lines 380-383).
After that ( lines 383-385), I dont understand why you repeated the same thing. I suppose it is a mistake.
Author Response
October 26, 2019
Dear Editor/Reviewers
We are pleased to resubmit the manuscript no. Plants-614825 (Round 2) entitled “Biological Control of Alternaria Leaf Spot Caused by Alternaria spp. in Jerusalem artichoke (Helianthus tuberosus L.) under Two Fertilization Regimes” after revision for re-evaluation and possible publication in the Plants.
The authors would like to thank the editor and reviewers for all valuable comments and suggestion on the manuscript in order to improve the manuscript.
We have revised the manuscript based on editor and reviewers, comments and suggestions. All of the points and responses from the editor and reviewers done and attached with this file.
Best Regards,
Sanun Jogloy

This manuscript is a resubmission of an earlier submission. The following is a list of the peer review reports and author responses from that submission.
Round 1
Reviewer 1 Report
Dear authors,
The general concept of your work and the context of the manuscript could be very interesting and helpful for the relevant readers but your work or at least the presented manuscript has some major issues that need to be addressed by major text revisions and probably additional experiments.
My specific comments are as follows:
Line 40: functionality of inulin in foods is due to her role as prebiotic and not as not digestible sweetener. Since fertilization is a major component of your experimentation this should be addressed in the abstract, the introduction (including the role of low and high fertilization schemes in plant disease development and especially Alternaria blight. Potentially the title should contain this as well. If experimental plots were in close proximity no separate soil analysis was needed and more importantly to be fully reported in the manuscript. The text requires to be written in a more clear and condensed way. It is quite difficult to follow and has repetitive parts as well as contradictory. Enzyme activity assay is fine but a more accurate approach for quantitative purposes can be nowadays employed (ie expression study/real time RT PCR). Figures depicting symptoms of plants are absolutely needed. Some tables can be omitted and be supplementary (such as 1, 2, 8, 11) Quantity (final concentration) of fungicide is not reported. One year trials with natural infection is barely acceptable for field trials. 10 days difference in planting date between the two fertilization regimes is not justified and no comment of its potential role to results. Fertilizers are applied and not "broadcasted" I am not sure what you mean by yield and yield components. Yield is an appropriate and inclusive term of all your measurements. Conclusions should be more elaborate. First of all the control of disease is good and the fact that Trichoderma is similar to fungicide is good. On this you should further your analysis and report on the cost of application and while considering all the benefits of the biocontrol application report on a cost/benefit analysis.I suggest major revision to your manuscript assuming you have available some of the required material/information.
Author Response
August 26, 2019
Dear Editor/Reviewers
We are pleased to resubmit the manuscript no Plants-566083 entitled " Biological Control of Alternaria Leaf Spot Caused by Alternaria spp. in Jerusalem artichoke (Helianthus tuberosus L.) under Two Fertilization Regimes" after revision for re-evaluation and possible publication in the Plants.
The authors would like to thank the editor and reviewers for all valuable comments and suggestion on the manuscript in order to improve the manuscript.
We did revise the manuscript base on editor and reviewers' comments and suggestions all of the points and responses to the editor and reviewers of each point had been done and attached with this file.
Best Regards,
Sanun Jogloy

Reviewer 2 Report
Although the topic can be considered of some interest, the article is very confusing. Authors present a huge amount of data, most of them useless. For instance, all information about the weather condition of soil composition is not interesting at all. It is necessary to reach table 12 and 13 to see some data on the main aim of the article, i.e. evaluating the effectiveness of Trichoderma or chemicals for controlling the disease.
Below a few more comments.
The introduction is very modest, with few or no details on the choice of the control methods, the origin and effectiveness of T9. The presentation of the results is really very hard to be followed. Authors do not put in evidence the "real" results (very modest) but rather present supposed statistically significant data. It is hard to believe that 10% (or less) disease incidence can be interesting from a practical point of view, or can be considered a real scientific advancement, even if statistically supported. Most of the tables should be "cleaned" and many can be deleted. Material and methods are not properly presented. Discussion is very modest, too: there should be many other aspects to be discussed, such as survival of the fungus, strategies for application, susceptibility to chemicals, just to make a few examples. The conclusion is just a repetition of what already said.
Author Response

(The authors gave the same response as above.)

Reviewer 3 Report
Trichoderma has many good qualities and useful application for agriculture. This manuscript is more proof that this organism can be fruitfully employed in a diverse set of agricultural situations. The detail of the data measurements and statistical analyses is impressive. And the manuscript is written with enough detail and explanation to tell a complete story. The 'flow' of this story begins to deteriorate a bit in the discussion sections. I believe all the pertinent information is present and appropriately-used references help support the points, but the wording becomes a little difficult to follow. To be specific - the wording was sometimes confusing with regards to referencing other studies and when referencing your own findings. I have made some suggestions for this section - but more work may still be needed for clarity. I would recommend a thorough readthrough of that section after all changes have been made. I have also made note of some changes that should be made and listed them below. Some of these changes are merely suggestive - to enhance the clarity for the reader.
Line 23 - Reposition the word "sprays" after the word "fungicide" (...and fungicide sprays...).
Line 39-40 - Rewrite as "...be used as a sweetener but cannot be digested by humans...".
Line 54 - Add the word "of" after the word "control".
Line 90 - Remove the word "in".
Line 104 - Replace the space in front of the colon to a position just after the colon.
Line 122 - Italicize "Trichoderma".
Line 123 - Should be rewritten in 1 of 2 ways, either "...letters had no significant..." or "...letters were not significantly different...". Also, this error should be corrected in the following lines: 125, 131, 205, 207, 215, 223, 225, 232, 234, 242, 279, 281, 306, and 310.
Line 153 - Remove the plural "s" from the word "methods".
Line 164 - This is a long sentence, but it needs one change to make the meaning clear - add the word "if" before the word "significant" (in line 164). Check to make sure the meaning still fits the intended purpose.
Line 189 - Remove the space before the end-of-sentence period.
Line 240 - Italicize "Trichoderma".
Line 317 - Italicize "viride".
Line 318 - Change "increase" to "increased".
Line 318 - Rewrite as "Trichoderma utilizes several mechanisms including antibiosis, competition, endophyte colonization,..."
Line 329 - Reword as "...chitinase decreased in both fertilization regimes because both plants were in fast growth and developmental stages...".
Line 334 - Rewrite as "...priming cucumber roots with Trichoderma spp..."
Line 337 - Rewrite as "...fungal infections in at least...".
Line 342 - Rewrite as "However, enzyme production may not be the only mechanism, mycoparisitism as seen by Yedidia et al. (35) may also help to control..."
Line 343 - Rewrite as "After spraying conidia, Trichoderma were able to colonize leaves of...".
Line 344 - Rewrite as "In another study it was show that, after foliar spray...".
Line 346 - Change "pathogen" to "pathogenic".
Line 355 - Rewrite as "...caused by Sclerotium rolfsii was reduced by...".
Line 361 - Remove the word "than high".
Line 363 - Rewrite "components" as "component".
Line 375-376 - Use subscript to write chemical formulas as is done in line 392.
Line 376 - Change "regimes" to "regime".
Line 377 - Change "10-days" to "10-day".
Line 384-385 - Rewrite as "...was then incubated for 10 days until green spores covered all seeds."
Line 390 - Remove the word "with".
Line 420 - Use superscript (exponent) for 10-7.
Line 448 - Use subscript for H2O.
Line 466 - Remove the word "amended".
Line 467 - Make all plants plural - the last one is singular.
Line 560 - Rewrite as "Minneapolis".
Author Response

(The authors gave the same response as above.)

Round 2
Reviewer 1 Report
Please provide your manuscript to a proficient person in English for a final review and editing in syntax and expressions.
Reviewer 2 Report
-